

# Reassembly and co-crystallization of a family 9 processive endoglucanase from its component parts: structural and functional significance of the intermodular linker

Svetlana Petkun[1], Inna Rozman Grinberg[1], Raphael Lamed[1], Sadanari Jindou[2], Tal Burstein[1], Oren Yaniv[1], Yuval Shoham[3], Linda J.W. Shimon[4], Edward A. Bayer[5] and Felix Frolow[1]

[1] Department of Molecular Microbiology and Biotechnology, The Daniella Rich Institute for Structural Biology, Tel Aviv University, Ramat Aviv, Israel
[2] Department of Life Sciences, Meijo University, Nagoya, Japan
[3] Department of Biotechnology and Food Engineering, Technion-Israel Institute of Technology, Haifa, Israel
[4] Department of Chemical Research Support, The Weizmann Institute of Science, Rehovot, Israel
[5] Department of Biological Chemistry, The Weizmann Institute of Science, Rehovot, Israel

Corresponding author
Edward A. Bayer,
ed.bayer@weizmann.ac.il

## ABSTRACT

Non-cellulosomal processive endoglucanase 9I (Cel9I) from *Clostridium thermocellum* is a modular protein, consisting of a family-9 glycoside hydrolase (GH9) catalytic module and two family-3 carbohydrate-binding modules (CBM3c and CBM3b), separated by linker regions. GH9 does not show cellulase activity when expressed without CBM3c and CBM3b and the presence of the CBM3c was previously shown to be essential for endoglucanase activity. Physical reassociation of independently expressed GH9 and CBM3c modules (containing linker sequences) restored 60–70% of the intact Cel9I endocellulase activity. However, the mechanism responsible for recovery of activity remained unclear. In this work we independently expressed recombinant GH9 and CBM3c with and without their interconnecting linker in *Escherichia coli*. We crystallized and determined the molecular structure of the GH9/linker-CBM3c heterodimer at a resolution of 1.68 Å to understand the functional and structural importance of the mutual spatial orientation of the modules and the role of the interconnecting linker during their re-association. Enzyme activity assays and isothermal titration calorimetry were performed to study and compare the effect of the linker on the re-association. The results indicated that reassembly of the modules could also occur without the linker, albeit with only very low recovery of endoglucanase activity. We propose that the linker regions in the GH9/CBM3c endoglucanases are important for spatial organization and fixation of the modules into functional enzymes.

## INTRODUCTION

Cellulose is a major component of the plant cell wall, lending structural stability and resilience to an otherwise flaccid material. The propensity of cellulose to form ordered, tightly packed, para-crystalline fibrils hinders its enzymatic degradation. Indeed, the recalcitrant properties of cellulose are such that numerous enzymes are required to act synergistically in achieving its efficient degradation. Many types of bacteria and fungi are capable of degrading cellulose and other plant cell wall polysaccharides in an effective manner, producing a variety of various cellulases and related enzymes, either existing in the free state, or associated with a multi-enzyme complex known as the cellulosome (*Bayer et al., 2004*; *Bayer et al., 2008*; *Demain, Newcomb & Wu, 2005*; *Doi & Kosugi, 2004*; *Fontes & Gilbert, 2010*). *Clostridium thermocellum* is an anaerobic thermophilic bacterium, known for its efficient degradation of cellulose and other plant cell wall polysaccharides (*Béguin, Millet & Aubert, 1992*; *Freier, Mothershed & Wiegel, 1988*; *Garcia-Martinez et al., 1980*; *Ng, Weimer & Zeikus, 1977*; *Wiegel, Mothershed & Puls, 1985*). The cellulase system of this bacterium includes a remarkable variety of enzymes, some existing in the free state but most associated with a highly efficient multi-enzyme complex, termed cellulosome (*Bayer et al., 2004*; *Lamed, Setter & Bayer, 1983*; *Lamed et al., 1983*; *Shoham, Lamed & Bayer, 1999*), capable of converting a wide variety of plant-derived polysaccharides directly into soluble sugars and fermentation products (*Béguin & Alzari, 1998*; *Felix & Ljungdahl, 1993*; *Schwarz, 2001*; *Schwarz, Zverlov & Bahl, 2004*). These capabilities render *C. thermocellum* a high utility candidate for use in consolidated bioprocessing (CBP) applications (reviewed in *Akinosho et al., 2014*).

Cellulases are a class of modular enzymes with a catalytic glycoside hydrolase (GH) module that hydrolyzes the $\beta$-1,4-glucosidic bond of the cellulose chain (*Cantarel et al., 2009*; *Davies & Henrissat, 1995*; *Gilbert & Hazlewood, 1993*; *Henrissat, 1991*; *Henrissat & Davies, 1997*; *Wilson & Irwin, 1999*). The catalytic module is usually associated with various numbers of accessory modules that serve to modulate the enzyme activity, and the enzymes have been categorized into families according to the amino-acid sequence of the GH domain (*Cantarel et al., 2009*; *Gilkes et al., 1991*; *Henrissat & Davies, 1997*; *Henrissat & Davies, 2000*; *Henrissat & Romeu, 1995*). Cellulases have been broadly divided into two types: endoglucanases that can hydrolyze bonds internally in cellulose chain, and exoglucanases that act preferentially on chain ends, progressively cleaving off cellobiose as the main product. The distinction between endo- and exo-acting enzymes is also reflected by the architecture of the respective class of active site, whereby endoglucanases, for example, are commonly characterized by a groove or open binding cleft, into which any part of the linear cellulose chain can fit. On the other hand, the exoglucanases bear tunnel-like active sites, which can only accept a substrate chain via its terminus (either the reducing or non-reducing end, depending on the enzyme), thereby cleaving cellulose in a sequential manner. The sequential hydrolysis of a cellulose chain has earned the term "processivity" (*Beckham et al., 2014*; *Davies & Henrissat, 1995*; *Wilson & Kostylev, 2012*), and processive enzymes are considered to be key components which contribute to the overall efficiency of a given cellulase system. Some endoglucanases, notably from GH family 9, have also

been shown to sequentially hydrolyze cellulose chains and are thus referred to as processive endoglucanases (*Gal et al., 1997*; *Gilad et al., 2003*; *Irwin et al., 1998*; *Jeon et al., 2012*; *Kuusk, Sorlie & Valjamae, 2015*; *Zverlov, Velikodvorskaya & Schwarz, 2003*). Such enzymes appear to possess extended catalytic clefts and the observed processivity appears to require highly coordinated substrate-binding affinities from opposite sides of the cleavage site (*Bu et al., 2012*; *Li, Irwin & Wilson, 2010*; *Payne et al., 2011*).

Cellulase 9I (Cel9I), is a non-cellulosomal family 9 processive endoglucanase from *Clostridium thermocellum*, which degrades crystalline cellulose (Avicel and filter paper) as well as phosphoric acid-swollen cellulose (PASC) and carboxymethyl cellulose (CMC) (*Gilad et al., 2003*). This enzyme consists of a catalytic GH9 module at its N terminus, followed by two family 3 carbohydrate-binding modules (CBMs): CBM3c and CBM3b. The three modules are separated by distinctive linker sequences. Such intermodular linker segments were proposed to be important for the physical association of the modules in the space, and to promote intermodular and/or intersubunit protein–protein interactions (*Bayer et al., 1998*; *Bayer et al., 2009*; *Noach et al., 2008*).

The C-terminal CBM3b module, as a classic CBM3, is responsible for targeting the Cel9I enzyme to the planar surface of the crystalline cellulose substrate (*Gilad et al., 2003*; *Su, Mackie & Cann, 2012*; *Tormo et al., 1996*). It has also been proposed to disrupt the crystalline regions of cellulose, rendering it more accessible to the GH9 catalytic module (*Yi et al., 2013*) and to contribute to enzyme processivity by preventing the desorption of the catalytic module from cellulose (*Telke et al., 2012*). The function of the CBM3c is less straightforward. Removal of CBM3c from *C. thermocellum* Cel9I, *C. cellulolyticum* Cel9G and *P. Barcinonensis* Cel9B significantly reduces the enzyme activity (*Burstein et al., 2009*; *Chiriac et al., 2010*; *Gal et al., 1997*). CBM3c modules have been shown to alter the normal function of the GH9 catalytic module of *Thermobifida fusca* Cel9A from the standard endo-acting mode into a processive endoglucanase (*Bayer et al., 1998*; *Irwin et al., 1998*). Thus, Gilad et al., showed in *2003* that the endoglucanase activity of Cel9I is dependent upon the presence of the CBM3c module and suggested that the fused CBM3c serves an important accessory role for the catalytic domain by altering its character to facilitate processive cleavage of recalcitrant cellulose substrates.

In addition to the Cel9 CBM3c, several other examples of CBMs that are considered to modulate catalytic specificity and act cooperatively with the catalytic domain have recently been discovered. These include CBM66 that directs the cognate enzyme towards highly branched glucans rather than linear fructose polymers (*Cuskin et al., 2012*), CBM48 that contributes to substrate binding at the active site of a glucan phosphatase (*Meekins et al., 2014*), family-43 $\beta$-xylosidases where the GH43 is complemented by an additional module that confers hydrolytic activity to the mature enzyme (*Moraïs et al., 2012*), and CBM46, that constitutes part of the catalytic cleft required for the hydrolysis of $\beta$-1,3-1,4-glucans (*Venditto et al., 2015*). The carbohydrate-binding PA14 domain is also known to affect substrate binding of the catalytic domain by contributing to the formation of its active site (*Gruninger et al., 2014*; *Zmudka, Thoden & Holden, 2013*).

We have previously shown that independently expressed GH9 and linker-containing CBM3c modules of Cel9I readily re-associate *in vitro* and that this physical reassociation recovers 60–70% of the intact Cel9I endoglucanase activity (*Burstein et al., 2009*).

We have examined in this work the interaction of the CBM3c with the catalytic module either with or without the intermodular linker in order to better understand the function of the CBM3c in the family-9 enzymes and the role of the linkers regions. The effect of the re-association of the CBM3c with linker (CBM3c*L*) and the CBM3c without linker (CBM3c*NL*) on the enzymatic activity of GH9 has been studied by the crystallization and structure determination of the reassembled GH9-CBM3c*L* complex at a resolution of 1.68 Å. The results of this study will help us to understand the contribution of ancillary modules in the action of multi-modular glycoside hydrolases.

## MATERIALS AND METHODS

### Cloning of the GH9, CBM3c*L* and CBM3c*NL* proteins

Cloning of the DNA fragments encoding the C-terminally His-tagged CBM3c with the linker and the untagged GH9 module from Cel9I of *C. thermocellum* (GenBank accession code L04735) was described earlier (*Burstein et al., 2009*; *Gilad et al., 2003*). C-terminally His-tagged CBM3c without the linker connecting it to the GH9 was amplified using the same procedure and the following primers: F′-5′CCATGGGCGAAGTTCCGGAGGATGAAATA and R′-5′CTCGAGCGGTTCCCTTCCAAATACCAG. The PCR products were purified and cleaved with restriction enzymes *Nco*I and *Xho*I and inserted into the pET-28a(+) expression vector (Novagen, Madison, WI, USA).

### Expression and purification of recombinant proteins

The GH9 and CBM3c modules both with (GH9L, CBM3c*L*) and without (GH9*NL* and CBM3c*NL*) the linker regions were expressed independently by the identical expression procedure. *Escherichia coli* strain BL21(DE3)RIL harboring the plasmids was aerated at 310 K in 3-liters Terrific Broth supplemented with 25 mg ml$^{-1}$ kanamycin. After 3 h, the culture reached an $A_{600}$ of 0.6; 0.1 mM isopropyl-$\beta$-D-1-thiogalactopyranoside was added to induce gene expression, and cultivation was continued at 310 K for an additional 12 h. Cells were harvested by centrifugation (5,000 × g for 15 min) at 277 K and were subsequently re-suspended in 50 mM NaH$_2$PO$_4$, pH 8.0, containing 300 mM NaCl at a ratio of 1 g wet pellet to 4 ml buffer solution. A few micrograms of DNase powder were added prior to the sonication procedure. The suspension was kept on ice during sonication, after which it was centrifuged (20,000 × g at 277 K for 20 min), and the supernatant was collected.

The soluble expressed His-tagged CBM3c modules with or without the linker, according to the type of the experiment, were applied batchwise to Ni-IDA resin during 1-h incubation with gentle stirring at 4 °C. Non-specifically bound proteins were washed with a buffer containing 50 mM NaH$_2$PO$_4$ pH 6, 300 mM NaCl, 10% glycerol and 10 mM imidazole. Crude extract supernatant fluids, containing the expressed GH9 module, were added to the CBM3c-bound Ni-IDA resin, and the mixture was incubated overnight with gentle stirring at 4 °C. The adsorbed protein complexes were eluted with 300 mM imidazole and

subjected to further purification by size-exclusion chromatography. Fast protein liquid chromatography (FPLC) was performed using a Superdex 75pg column and ÄKTA Prime system (GE Healthcare, Piscataway, New Jersey, USA) to further purify the complex. One peak, corresponding approximately to 70 kDa, matching the predicted molecular weight of the GH9-CBM3c complex, was observed in the chromatogram. The 15 amino-acid linker sequence (about 1.5 kDa) did not significantly affect the elution volume, compared to that of the complex without the linker, presumably due to the limited resolution of the column. The relevant fractions (the purified complexed proteins) were analyzed by 15% sodium dodecyl sulfate-polyacrylamide gel electrophoresis (SDS-PAGE) with Coomassie brilliant blue staining. Two clear bands, of about 52 and 19.5 kDa were observed. The rearranged modules were concentrated to 6 mg ml$^{-1}$ using Centriprep YM-3 centrifugal filter devices YM-3 (Amicon Bioseparation, Millipore Corporation, Bedford, Massachusetts, USA). Protein concentration was determined by measuring UV absorbance at 280 nm.

The full-length Cel9I was purified by affinity chromatography on Avicel as reported earlier (*Burstein et al., 2009*; *Gilad et al., 2003*).

## Microcalorimetric analysis

Isothermal titration calorimetry (ITC) experiments were carried out using a VP-ITC MicroCalorimeter (MicroCal, LLC, Northampton, Massachusetts, USA) at 298 K. About 300 μM solution of CBM3c*NL* was injected into a 65 μM solution of GH9. The reaction was performed in a buffer containing 50 mM Tris–HCl, pH 7.5, 150 mM NaCl, 0.05% sodium azide. Heats of dilution of the titrants were subtracted from the titration data, and the corrected data were analyzed using the Origin ITC analysis software package supplied by MicroCal. Thermal titration data were fit to the one binding site model, and enthalpy (ΔH), entropy (ΔS), association constant (Ka) and stoichiometry of binding (N) were determined. In all cases, the calculated stoichiometry (N) was lower than one, most likely due to the fact that the CBM3 proteins lost their native functionality with time. For the analysis, the CBM3 protein concentrations were corrected as to provide a stoichiometry of one. Two titrations were performed to evaluate reproducibility.

## Enzyme activity assay

Reactions were performed at 333 K, in 50 mM citrate buffer (pH 6.0). The soluble cellulolytic substrate was carboxymethyl cellulose (CMC, Sigma Chem. Co. St. Louis, Missouri, USA). The amount of reducing sugars released from the substrate was determined with the 3,5-dinitrosalicylic acid (DNS) reagent as described by Miller et al. (*Miller, 1959*). Activity was defined as the amount (micromole) of reducing sugar released after 10 min of reaction.

## Crystallization

Initially the protein samples containing 6 mg/ml protein solution in 1.2 mM Tris–HCl pH 7.5, 1.5 mM sodium chloride, 0.025% sodium azide, were screened, using the microbatch crystallization method under 1:1 mixture of silicon and paraffin oil (*Chayen et al., 1990*), using 288 conditions from the Hampton Research HT screens (SaltRx, Index HT, and Crystal Screen HT; Hampton Research, Aliso Viejo, California, USA) and 96 conditions

of the Wizard Crystallization kit from Emerald BioSystems (Rigaku Reagents, Bainbridge Island, Washington, USA). The dyad of GH9 and CBM3c*NL* did not yield any crystals. Screening of the GH9-CBM3c*L* resulted in plate-like crystals that appeared after several days under several conditions, all of which contained PEG 3350 and 4000. The best crystals were obtained in 30% PEG (both 3350 and 4000), 0.2 M magnesium chloride, and 0.1 M Hepes, pH 7.5. Attempts to optimize this condition using microbatch, hanging-drop, and sitting drop methods were unsuccessful, as the crystals remained very thin and fragile. The superfine Eyelash (Ted Pella, Inc, Redding, California, USA) was used to touch these crystals and consequently to streak the sitting drops, composed of 5 μl of the protein solution and 5 μl of the precipitating solution (24% PEG 3350, 0.2 M magnesium chloride, 0.1 M Hepes, pH 8.0). After one day, crystals of different morphology, with maximum size of about 0.05 mm, appeared in the drop.

## Data collection and crystallographic analysis

The crystals of the GH9-CBM3c*L* complex were harvested from the crystallization drop using a nylon cryo-loop (Hampton Research, Aliso Viejo, California, USA). For data collection, crystals were mounted on the MiTeGen stiff micro-mount (MiTeGen, Ithaca, New York, USA) made of polyimide and flash-cooled in a nitrogen stream produced by Oxford Cryostream low temperature generator (*Cosier & Glazer, 1986*) at a temperature of 100 K. Mother-liquor of the crystals served for cryo-protection during the cooling in liquid nitrogen.

Diffraction data from the GH9-CBM3c*L* crystals were measured using the ID23-2 beam line at ESRF, Grenoble, France. A MAR CCD 225 area detector and X-ray radiation of 0.873 Å wavelength were used. Diffraction data of 480 images with 0.5°oscillation per image were collected. Data were processed with *DENZO* and scaled with *SCALEPACK* as implemented in *HKL2000* (*Otwinowski & Minor, 1997*). The crystals diffracted to 1.68 Å resolution and belong to the orthorhombic space group P$2_1 2_1 2_1$, with unit cell parameters $a = 70.4$, $b = 88.5$, $c = 106.5$ Å. There is one GH9-CBM3c*L* complex per asymmetric unit with a Matthews density $V_M$ of 2.37 Å$^3$ Da$^{-1}$, corresponding to a solvent content of 48.15% (*Matthews, 1968*). The X-ray data analysis statistics are presented in Table 1 (*Stout & Jensen, 1968*).

Molecular replacement was carried out with *MOLREP* (*Vagin & Teplyakov, 1997*), using the coordinates of the GH9 and CBM3c modules of endoglucanase 9G from *Clostridium cellulolyticum* (PDB code 1G87, 66 and 51% sequence identity, respectively), as a search model. The *MOLREP* calculations with the GH9 domain converged into a clear solution with 1 molecule in the asymmetric unit with an R-factor of 0.533 and correlation coefficient of 0.567. This solution was inserted into *MOLREP* calculations as a fixed molecule and the coordinates of CBM3c module were used for the search producing a solution with an $R_{cryst}$ of 0.505, and correlation coefficient of 0.582. The resulting model with 5% of reflections forming test set (*Brünger, 1992*) was subjected to 10 cycles of restrained refinement using anisotropic B-factors, yielding the $R_{cryst}$ and $R_{free}$ 0.329 and 0.359, respectively (*REFMAC5*) (*Murshudov, Vagin & Dodson, 1997*). Automated model

**Table 1** **Diffraction data of the GH9-CBM3c *in vitro* reassembled complex from Cel9I from *C. thermocellum*.** Values shown in parentheses are for the highest resolution cell.

| | |
|---|---|
| Space group | $P2_12_12_1$ |
| Number of crystals | 1 |
| Total rotation range (°) | 240 |
| $a$ (Å) | 70.39 |
| $b$ (Å) | 88.54 |
| $c$ (Å) | 106.49 |
| $V$ (Å$^3$) | 663,743.40 |
| Resolution range (Å) | 30–1.68 (1.71–1.68) |
| Total number of reflections | 676,571 |
| Unique reflections | 76,727 |
| Mosaicity range (°) | 0.18–0.46 |
| Average redundancy | 9.0 |
| Completeness, overall (%) | 97.9 (74.8) |
| Mean I/$\sigma$ (I) | 34.72 (2.08) |
| $R_{\mathrm{merge}}$[a] (%) | 7.4 (49.8) |

**Notes.**

[a] $R_{\mathrm{merge}} = \Sigma_{hkl}\Sigma_i |I_i(hkl) - \langle I(hkl)\rangle| / \Sigma_{hkl}\Sigma_i I_i(hkl)$, where $\Sigma_{hkl}$ denotes the sum over all reflections and $\Sigma_i$ the sum over all equivalent and symmetry-related reflections.

building by *ARP/wARP* (*Perrakis, Morris & Lamzin, 1999*) produced a complete structure with $R_{\mathrm{cryst}}$ and $R_{\mathrm{free}}$ of 0.218 and 0.243, respectively. The model was manually corrected using *COOT* (*Emsley & Cowtan, 2004*) and refined using *REFMAC5* (*Murshudov, Vagin & Dodson, 1997*). The $R_{\mathrm{cryst}}$ and $R_{\mathrm{free}}$ improved to 0.184 and 0.228, respectively. Solvent atoms were built using *ARP/warp* (*Perrakis, Morris & Lamzin, 1999*). Refinement of TLS (rigid body translation/libration/screw motions) parameters was performed (*Winn, Isupov & Murshudov, 2001*; *Winn, Murshudov & Papiz, 2003*). The model was subjected to several additional cycles of manual rebuilding and refinement. The model converged to final $R_{\mathrm{cryst}}$ and $R_{\mathrm{free}}$ factors of 0.144 and 0.176, respectively.

The refinement statistics of the structure are summarized in Table 2. The structure was validated using *MolProbity* (*Davis et al., 2007*).

## Protein sequence analysis

Sequence alignments were performed using *CLUSTALW* (*Larkin et al., 2007*) and the coloring of residues (representing degree of conservation) using ProtSkin (*Deprez et al., 2005*). Sources of the sequences used in this work are as follows: *Clostridium thermocellum* Cel9I GH9 module, CBM3c and CBM3b (AAA20892.1); *Clostridium cellulolyticum* Cel9G GH9 module, CBM3c (AAA73868.1); *Thermobifida fusca* Cel9A GH9 module and CBM3c (AAB42155.1); *Cellulomonas fimi* Ce9A CBM3c (AAA23086.1); *Clostridium cellulovorans* EngH CBM3c (AAC38572.2) and CbpA CBM3a (AAA23218.1); *Clostridium stercorarium* CelZ CBM3c and CBM3b (CAA39010.1) and CelY CBM3b (CAA93280.1); *Clostridium thermocellum* CipA CBM3a (CAA48312.1), CelQ CBM3c (BAB33148.1), Cel9V CBM3c' and CBM3b' (CAK22315.1), Cel9U CBM3c' and CBM3b' (CAK22317.1) and Cbh9A CBM3b (CAA56918.1); *Clostridium cellulolyticum* CipC CBM3a (AAC28899.2) and CelJ

**Table 2** Refinement statistics and results of *MolProbity* validation for reassembled GH9-CBM3c complex.

| | |
|---|---|
| Space group | $P2_12_12_1$ |
| Resolution range | 30–1.68 |
| No. of reflections in working set | 71,559 |
| No. of reflections in test set | 3,580 |
| No. of protein atoms | 5,071 |
| No. of solvent atoms | 835 |
| No. of Cl ion atoms | 3 |
| No. of Ca ion atoms | 2 |
| Overall B factor from Wilson plot ($\text{Å}^2$) | 16.06 |
| Averaged B factor ($\text{Å}^2$) | 21.12 |
| $R_{cryst}$ | 0.1441 |
| $R_{free}$ | 0.1759 |
| **Geometry** | |
| RMS bonds (Å) | 0.014 |
| RMS bond angles (°) | 1.371 |
| **MolProbity validation** | |
| Ramachandran favored (%) (goal > 98%) | 96.7 |
| Ramachandran outliers (%) (goal < 0.2%) | 0.5 |
| $C_\beta$ deviations > 0.25 Å (goal 0) | 1 |
| Clash score[a] (all atoms) | 2.88 |
| Rotamer outliers (%) (goal < 1%) | 0.8 |
| Residues with bad bonds (%) (goal < 1%) | 0.00 |
| Residues with bad angles (%) (goal < 0.5) | 0.33 |

**Notes.**

[a] Clash score is the number of serious steric overlaps (>0.4 Å) per 1,000 atoms.

CBM3c (AAG45158.1); *Acetivibrio cellulolyticus* Cel9B CBM3c' and CBM3b' (CAI94607.1) and CipV (ScaA) CBM3b (AAF06064.1); *Clostridium josui* CipA (CipJ) CBM3a (BAA32429.1); *Bacteroides cellulosolvens* ScaA CBM3b (AAG01230.2); *Bacillus subtilis* CelA CBM3b (AAA22307.1); *Pectobacterium atrosepticum* CelVI CBM3b (X79241.2); *Bacillus licheniformis* CelA CBM3b (CAJ70714.1).

# RESULTS

## Cloning, expression and purification of Cel9I and its modular components

The full-length *C. thermocellum* Cel9I enzyme and its individual component parts were over-expressed in *Escherichia coli*, according to *Burstein et al. (2009)*, in order to investigate the contribution of the ancillary modules and their linkers to the catalytic activity of the enzyme. These include the isolated GH9 module with and without a His tag, the His-tagged CBM3c module together with its adjacent N-terminal linker that connects it to the GH9 module (CBM3c*L*) and His-tagged CBM3c module without the N-terminal linker (CBM3c*NL*). For details, see Fig. 1. Following purification procedures, all recombinant proteins showed a single band in SDS-PAGE of the anticipated molecular masses.

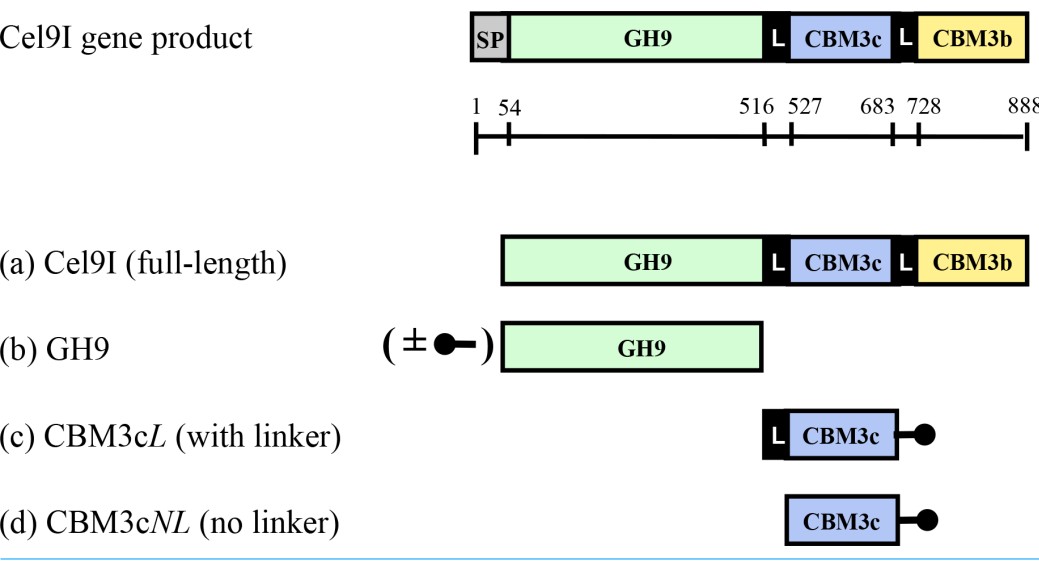

**Figure 1 Schematic diagram of the Cel9I gene product and the recombinant proteins (A–D) prepared for this study.** The GH9 module alone (B) was prepared with and without an N-terminal His tag (shown schematically in the figure), and the CBM3c's were prepared with C-terminal His tags. Scale shows the number of amino acid residues and the boundaries of the different regions of the protein.

## Recovery of endoglucanase activity upon association of CBM3c*NL* and GH9 compared to CBM3c*L* and GH9

Previous works (*Burstein et al., 2009*; *Gilad et al., 2003*) demonstrated that the Cel9I catalytic module alone has no detectable activity on CMC (carboxymethyl cellulose) and that adding the CBM3c*L* to form the Cel9I-CBM3c*L* dyad serves to recover up to 70% of the lost activity. To further examine the importance of the linker connecting the GH9 and the CBM3c modules, we tested the ability of CBM3c*NL* to recover the CMCase activity of GH9. A fixed amount of the catalytic module (70 pmol in 400 μl) was mixed with increasing amounts of CBM3c*L* or CBM3c*NL*. The activity of the intact Cel9I enzyme was defined as 100%, and the activity of the reconstituted complexes was measured relative to that of Cel9I. The results indicated that GH9-CBM3c*NL* exhibit only about 10% of the intact Cel9I activity towards CMC, whereas the reassembled GH9-CBM3c*L* provided up to 50% of the activity (Fig. 2). The fact that a higher than one molar ratio was required to obtain maximum activity can be explained by the fact that the CBM protein was only partly functional as was also observed in the ITC experiments described below. Overall, the results suggest that the linker is required for better fitting of the reconstituted CBM3c which results in better recovered activity.

## Overall structure of the reassembled GH9-CBM3c

Initially, we tried to overexpress and purify the covalently linked GH9-CBM3c, however the full-length protein was unstable and proteolysis occurred during the overexpression and purification stages, partially resulting in separate GH9 and CBM3c modules. Therefore, the obtained protein samples were not homogenous and were not suitable for crystallization trials. Instead, we employed an alternative approach where we expressed the

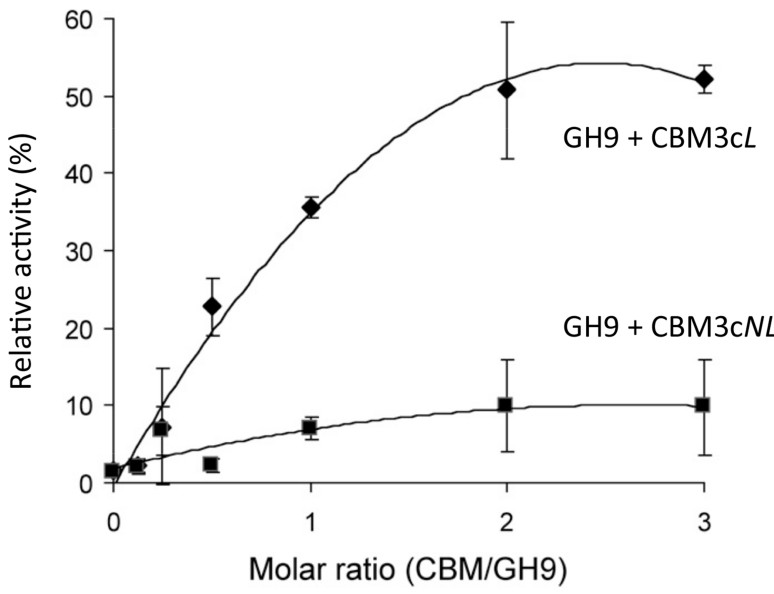

**Figure 2 Recovery of activity upon association of CBM3c (with and without linker) and GH9.** CMCase activity (μmol reducing sugar released in a 10-min reaction) of His-tagged GH9, mixed either with CBM3c*L* (diamonds) or CBM3c*NL* (squares), was examined. A fixed amount (70 pmol) of the GH9 catalytic module was mixed with increasing amounts of the indicated helper module, and their respective activities were compared to that of the intact Cel9I (GH9-CBM3c-CBM3b, set as 100%).

two domains separately and combined them *in vitro*. Surprisingly, the combined modules crystallized and formed a structure similar to those of the known GH9 cellulases.

The crystal structure of the reassembled *C. thermocellum* Cel9I GH9-CBM3c*L* dyad was determined by molecular replacement and the coordinates are deposited in Protein Data Bank with code 2XFG. Data collection and refinement statistics are given in Tables 1 and 2. The catalytic GH9 and the ancillary CBM3c modules reassembled *in vitro* to form a dyad (Fig. 3A) similar in structure to the intact tandem GH9-CBM3c modules of the orthologous endoglucanases: Cel9G from *C. cellulolyticum* (1G87) and Cel9A (previously termed cellulase E4) from *Thermobifida fusca* (1TF4), with an RMS deviation of 0.783 Å over 468 Cα atoms with Cel9G and 0.757 Å with Cel9A (Fig. 3B).

## Structure of the GH9 module

The catalytic module of the Cel9I enzyme consists of residues 1–446, comprising 15 $\alpha$-helices, whereby the twelve longest ones form the $(\alpha/\alpha)_6$-barrel (Fig. 4A). The hydrophobic core of the GH9 module is formed by 118 hydrophobic and aromatic amino acids, the vast majority of which are also conserved in the GH9 modules from *C. cellulolyticum* Cel9G and *T. fusca* Cel9A. Hydrophobic and aromatic cores have been proposed to play an important role in the formation of $(\alpha/\alpha)_6$-barrels (*Mandelman et al., 2003*). The GH9 module of Cel9I thus shows high structural similarity with the two latter GH9 structures: *C. cellulolyticum* Cel9G (0.367 Å RMS deviation over 349 C-alpha atoms) and *T. fusca* Cel9A (0.532 Å RMS deviation over 359 C-alpha atoms).

**A**

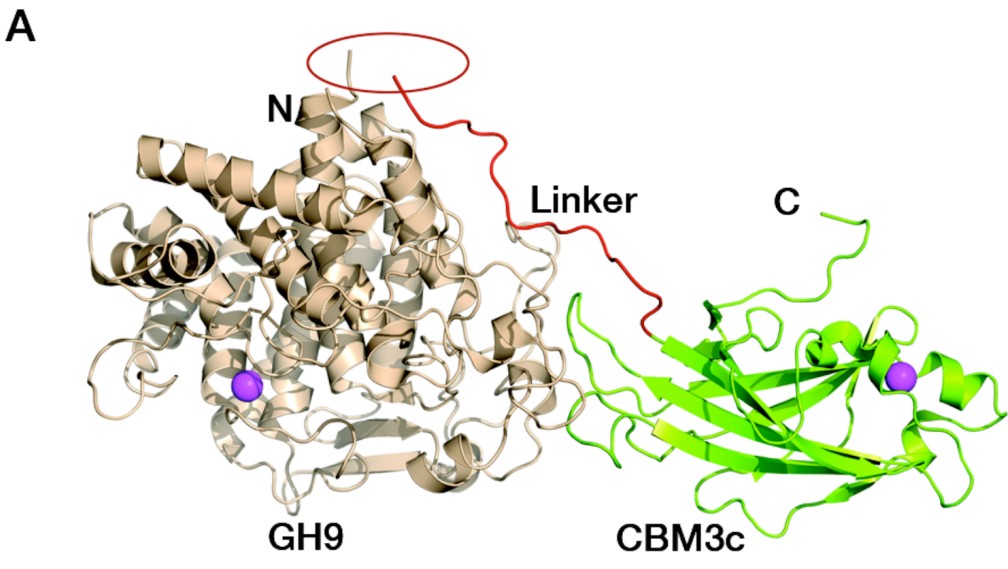

**B**

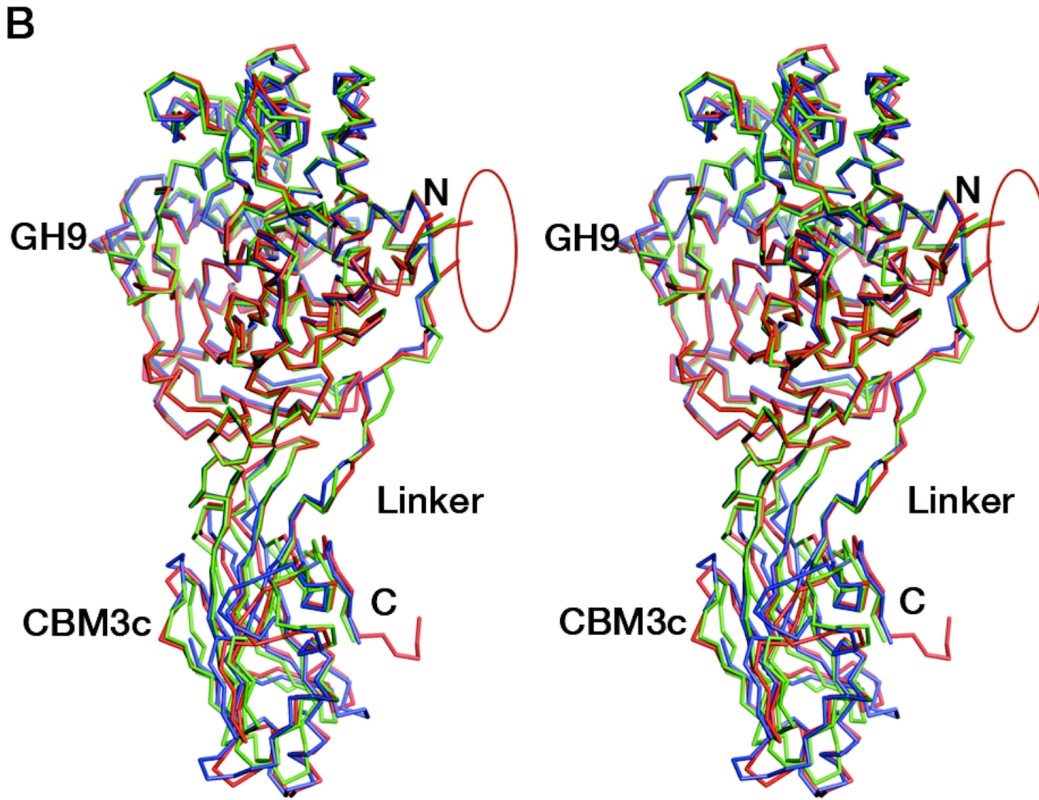

**Figure 3  Reassembled GH9-CBM3c from Cel9I.** C and N termini are indicated, and the break between the GH9 and CBM3c modules is marked by a red ellipse. (A) The *in vitro* reassembled complex of the catalytic (GH9, wheat) and carbohydrate-binding (CBM3c, green) modules of Cel9I from *C. thermocellum*, cartoon representation. Calcium atoms are shown as magenta-colored spheres. (B) Stereo-view (cross-eyed) of the superposition of the reassembled GH9-CBM3c structure of *C. thermocellum* Cel9I (red) with the bimodular structures of *C. cellulolyticum* Cel9G (blue) and *T. fusca* Cel9A (green).

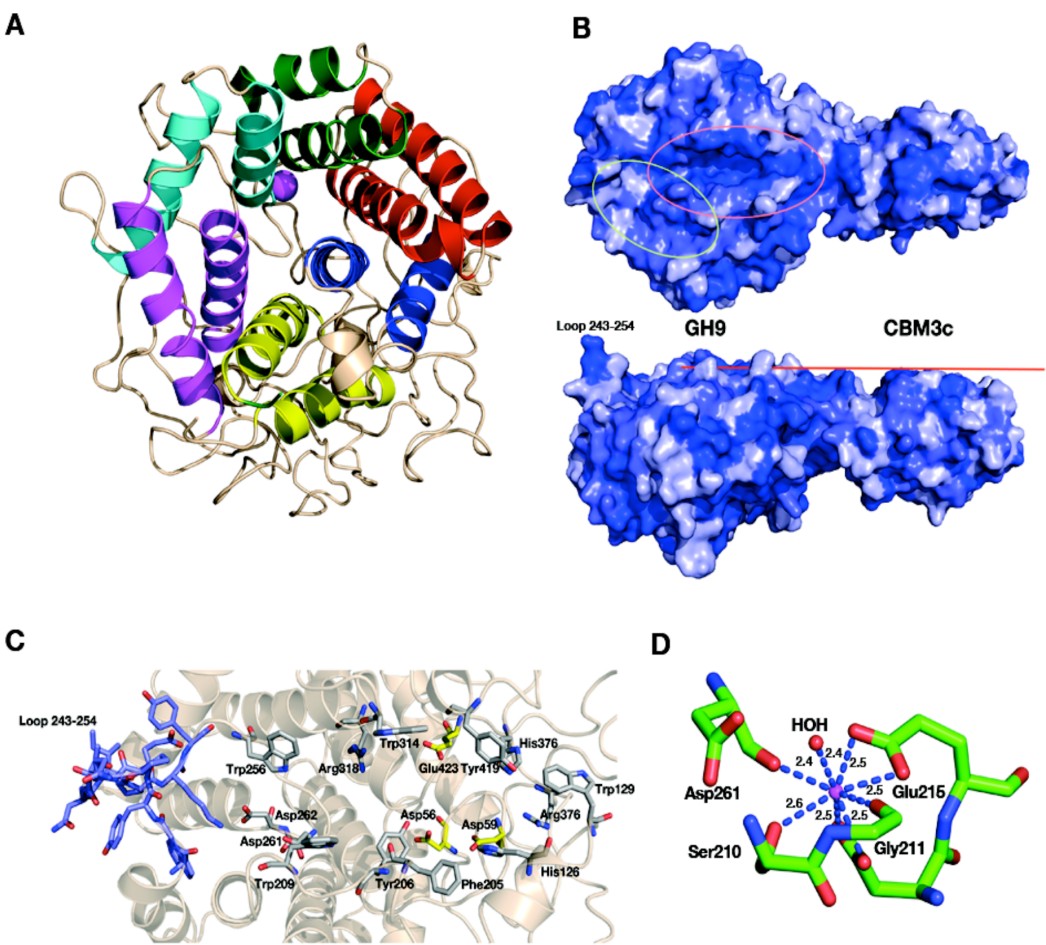

**Figure 4 Structural components of the reassembled *C. thermocellum* GH9-CBM3c.** (A) Structure of the GH9 catalytic module, cartoon representation. Twelve $\alpha$-helices form an $(\alpha/\alpha)_6$-barrel fold. Pairs of helices, comprising the fold, are emphasized by red, blue, yellow, magenta, cyan and green. (B) Surface representation of the reassembled GH9-CBM3c complex. The residues are shaded according to the extent of their conservation with Cel9G from *C. cellulolyticum* and Cel9A from *T. fusca*. Darker blue indicates higher conservation. Top, birds-eye view of the catalytic cleft. Bottom, lateral view, showing the flat surface (red bar). Pink ellipse indicates the catalytic cleft, and green ellipse designates terminal portion of the catalytic site. (C) Close-up (same orientation as in B, top) of the catalytic cleft of the Cel9I GH9 module showing functional residues. Carbohydrate-binding residue carbons are colored gray, catalytic residue carbons are colored yellow. Loop 243–254 carbons are colored in light blue. (D) Calcium-binding site of the *C. thermocellum* Cel9I GH9 module. Coordinating residues are shown in stick representation. The calcium ion is colored magenta, and distances to the coordinating atoms are indicated.

The catalytic site of the GH9 module is located at the depression in the flat surface, formed by the loops connecting the N termini of the barrel helices (Fig. 4B). The flat face is rich in charged and polar residues (Fig. 4B), highly conserved also in Cel9G (1G87) and Cel9A (1TF4). The GH9 modules of these cellulases (*Mandelman et al., 2003*; *Sakon et al., 1997*; *Zhou et al., 2004*) exhibit similar flat faces and clefts, and these conserved residues (His 126, Trp 129, Phe 205, Tyr 206, Trp 209, Trp 256, Asp 261, Asp 262, Trp 314, Arg 318, His 376, Arg 378, and Tyr 419) have been shown to bind natural and synthetic oligosaccharides (Fig. 4C). In the present structure, as in the other known GH9-CBM3c

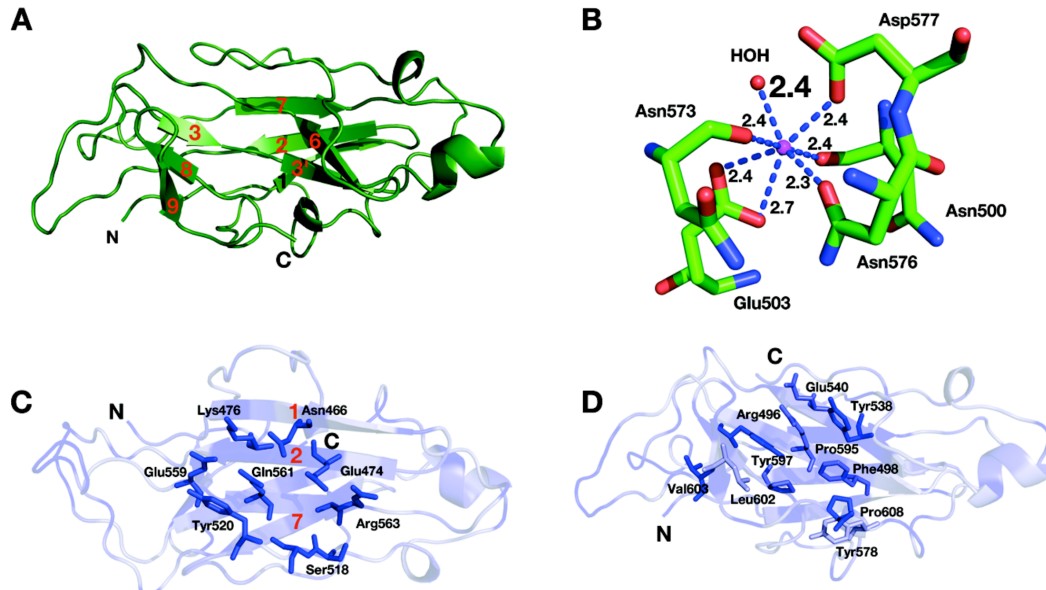

**Figure 5 Structure of the CBM3c of Cel9I from *C. thermocellum*.** C and N termini are indicated (A). Cartoon representation, β-strands are numbered according to the alignment with Cel9G from *C. cellulolyticum*, and Cel9A from *T. fusca*. (B) Calcium-binding site of the CBM3c. (C) Birds-eye view of the flat surface. Residues are shaded according to their degree of conservation with *C. cellulolyticum* Cel9G and *T. fusca* CEL9A. Surface-exposed conserved residues are shown in stick representation. (D) Shallow groove of the CBM3c. Conserved surface residues are shown in stick representation. The residues are colored according to the degree of conservation in CBM3a, CBM3b and CBM3c modules derived from the sequences listed in the Methods section.

bimodular structures, one end of this cleft is blocked by a loop formed by residues 243–254 and the other end is fused with the flat surface of the CBM3c module (Fig. 4B). Details of the catalytic cleft are presented in Fig. 4C.

One calcium ion is found near the catalytic cleft of the GH9 module of Cel9I and is coordinated by a Ser 210 (OG) 2.6 Å, Gly 211 (O) 2.4 Å, Asp 261 (O) 2.4 Å, Asp 214 bifurcated (OD1, OD2) 2.5 Å, and Glu 215 bifurcated (OD1, OD2) 2.5 Å (Fig. 4D). Despite some minor changes in the residues of coordination this ion seems to be structurally equivalent to those of *T. fusca* Cel9A (RMS deviation 0.160 Å over 5 Cα atoms of the coordinating residues), and *C. cellulolyticum* Cel9G (RMS deviation 0.503 Å over 4 Cα atoms). In all three cases the calcium ion draws together the N-terminal ends of α-helices 8 and 10.

## Structure of the CBM3c module

The CBM3c module consists of 150 amino acids arranged in an eight β-stranded sandwich motif homologous to other known family 3 CBM structures (*Gilbert, Knox & Boraston, 2013*; *Mandelman et al., 2003*; *Petkun et al., 2010b*; *Sakon et al., 1997*; *Shimon et al., 2000*; *Tormo et al., 1996*; *Yaniv et al., 2014*; *Yaniv et al., 2012b*; *Yaniv et al., 2011*). The "lower" face of the sandwich is formed by β-strands 1, 2, and 7; the "upper" face is formed by β-strands 3, 3′, 6, 8, and 9 (Fig. 5A). The structure of Cel9I CBM3c is particularly similar to the structures of the other two previously described CBM3c structures (RMS deviation 0.734 Å over 116 C-alpha atoms with CBM3c from *C. cellulolyticum* Cel9G; RMS deviation 0.829 Å

over 113 atoms with CBM3c from *T. fusca* Cel9A). Only 31% of amino acids are located in $\beta$-strands of the CBM3c module from Cel9I; others are found in the loop regions.

One calcium ion was found in the upper $\beta$-sheet of the CBM3c molecule (Fig. 5B) and is coordinated by a water molecule and five residues from the upper $\beta$-sheet: Asn 500 (O), Glu 503 bifurcated (OE1, OE2), Asn 573 (O), Asn 576 (OD1), Asp 577 (OD1). This calcium atom is in a similar location as in Cel9A and Cel9G, and probably plays a structural role for most CBM3 modules, as was suggested previously (*Tormo et al., 1996*).

The lower sheet forms a flat platform conserved between the CBM3c modules and the other two molecular structures. This flat surface is rich in charged and polar conserved surface residues: Asn 466, Glu 474, Lys 476, Ser 518, Tyr 520, Glu 559, Gln 561, and Arg 563 (Fig. 5C). The planar region of the CBM3c modules in all three enzymes is particularly aligned in continuation of the catalytic cleft of the catalytic modules, and has been proposed to bind single chains of cellulose and guide them to the cleft (*Mandelman et al., 2003*; *Sakon et al., 1997*).

The CBM3c possesses a very interesting surface structure, formed by the $\beta$-strands on the opposite side of the flat surface, called the "shallow groove" (*Shimon et al., 2000*; *Tormo et al., 1996*). The "shallow groove" is lined by four aromatic rings (Phe 498, Tyr 538, Tyr 578 and Tyr 597), two charged or polar residues (Arg 496, and Glu 540), Leu 602, Pro 595 and Pro 608. These residues are also conserved in other CBM3 modules regardless of their subgroup relation (a, b, or c), their cellulose-binding ability and their effect on the activity of the catalytic module. Figure 5D shows the shallow groove of the CBM3c module from the Cel9I enzyme colored according to the extent of the conservation of the residues in other CBM3a, b and c modules (darker blue represents more conservation). The alignment was performed over 25 CBM3 sequences (11 CBM3c, 12 CBM3b and CBM3b', and 4 CBM3a). Conservation of this surface structure, regardless of the particular known function of the CBMs, implies that this site has some kind of "generic" function. This shallow groove may serve to bind to single oligosaccharide chains or to peptide chains, such as the intermodular linkers common to cellulases or cellulosomal scaffoldin subunits. There is evidence that the shallow groove interacts with a linker region (*Petkun et al., 2010a*; *Shimon et al., 2000*; *Yaniv et al., 2012a*).

## Contact residues between the GH9, linker and CBM3c

The *in vitro* reassembled GH9-CBM3c*L* complex has a large intermodular interface, the contact area of which is 1,108.3 Å$^2$, corresponding to 12.3% of the total surface-exposed area of the CBM3c module and 6.2% of the exposed GH9 module (*Krissinel & Henrick, 2007*). The GH9 and the CBM3c*L* modules of Cel9I are assembled into the reconstituted GH9-CBM3c complex by 31 hydrogen bonds (4 main chain-main chain, 19 main chain-side chain, and 8 side chain-side chain), 14 hydrophobic, 3 aromatic interactions, and 3 ionic bonds (http://pic.mbu.iisc.ernet.in/index.html) (*Tina, Bhadra & Srinivasan, 2007*). Sixteen residues from the GH9 module and seventeen residues from the CBM3c participate in these interactions (contact residues are shown in Fig. 6A). The vast majority of the contact residues and contacts are similar to those of *C. cellulolyticum* Cel9G and of *T. fusca* Cel9A (Fig. 6B).

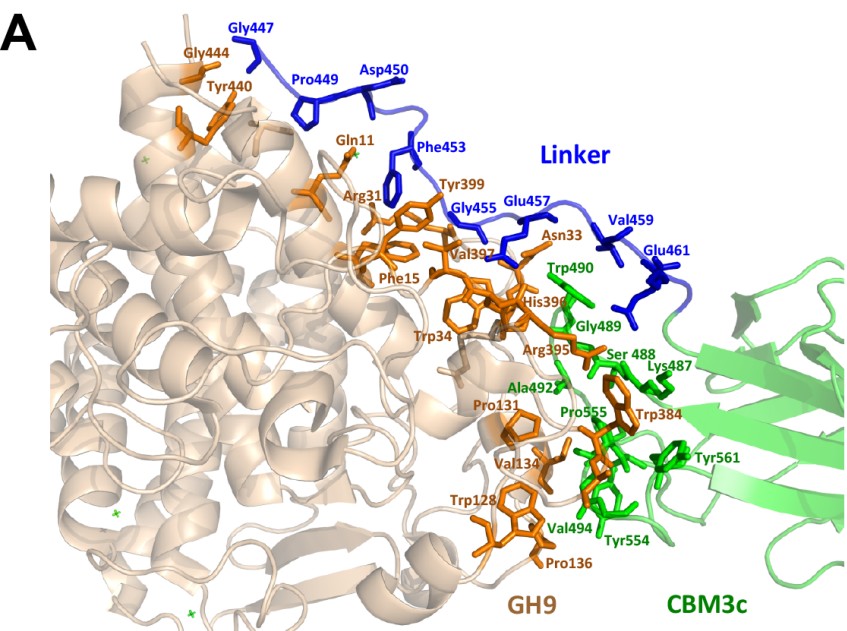

**B**

**GH9 module**

```
CelI    TGAFNYGEALQKAIFFYECQRSGKLDSSTLRLNWRGDSGLDDGKDAGIDLTGGWYDAGDH 60
CelG    AGTYNYGEALQKSIMFYEFQRSGDLPAD-KRDNWRDDSGMKDGSDVGVDLTGGWYDAGDH 59
E4      EPAFNYAEALQKSMFFYEAQRSGKLPEN-NRVSWRGDSGLNDGADVGLDLTGGWYDAGDH 59

        ------------------------------------------------------------

CelI    DGHADHAWWGPAEVMPMERPSYKVDRSSPGSTVVAETSAALAIASIIFKKVDGEYSKECL 180
CelG    DGGKDHSWWGPAEVMQMERPSFKVDASKPGSAVCASTAASLASAAVVFKSSDPTYAEKCI 179
E4      DGDADHKWWGPAEVMPMERPSFKVDPSCPGSDVAAETAAAMAASSIVFADDDPAYAATLV 179

        ------------------------------------------------------------

CelI    G--RSFVVGFGENPPKRPHHRTAHGSWADSQMEPPEHRHVLYGALVGGPDST-DNYTDDI 416
CelG    G--RSFVVGYGVNPPQHPHHRTAHGSWTDQMTSPTYHRHTIYGALVGGPDNA-DGYTDEI 413
E4      PRNSSYVVGFGNNPPRNPHHRTAHGSWTDSIASPAENRHVLYGALVGGPGSPNDAYTDDR 417

CelI    SNYTCNEVACDYNAGFVGLLAKMYKLYGEL    446
CelG    NNYVNNEIACDYNAGFTGALAKMYKHSG —   441
E4      QDYVANEVATDYNAGFSSALAMLVEEYG —   445
```

**CBM3c module**

Linker
```
CelI    GSPDPKFNGIEEVPEDEIFVEAGVNASGNNFIEIKAIVNNKSGWPARVCENLSFRYFINI 506
CelG    GDPIPNFKAIEKITNDEVIIKAGLNSTGPNYTEIKAVVYNQTGWPARVTDKISFKYFMDL 501
E4      GTPLADFPPTEEPDGPEIFVEAQINTPGTTFTEIKAMIRNQSGWPARMLDKGTFRYWFTL 505

CelI    EEIVNAGKSASDLQVSSSYNQGAKLS--DV--KHYKDNIYYVEVDLSGTKIYPGGQSAYK 562
CelG    SEIVAAGIDPLSLVTSSNYSEGKNTKVSGVLPWDVSNNVYYVNVDLTGENIYPGGQSACR 561
E4      DE----GVDPADITVSSAYNQCATPE--D--VHHVSGDLYYVEIDCTGEKIFPGGQSEHR 557

CelI    KEVQFRISAPEGTV-FNPENDYSYQGLSAGTV-VKSEYIPVYDAGVLVFGREPLE 615
CelG    REVQFRIAAPQGTTYWNPKNDFSYDGLPTTSTVNTVTNIPVYDNGVKVFGNEP-- 614
E4      REVQFRIAGGPG---WDPSNDWSFQGIGNE--LAPAPYIVLYDDGVPVWGTAP-- 605
```

**Figure 6  Contact residues of the reassembled GH9-CBM3c complex.** (A) The GH9 module is colored in brown, CBM3c in green and the linker in blue. Contact residues of the GH9, CBM3c and linker are colored orange, green and blue, respectively. The contact residues between the linker and the domains are described in the text. (B) Alignment of the GH9 and CBM3c modules of *C. thermocellum* Cel9I, *C. cellulolyticum* Cel9G, and *T. fusca* Cel9A (E4) cellulases. Contact residues are highlighted in yellow. Only the relevant regions of the alignment are shown. Residues of linker sequences are shown blue font.

Conserved residues of the linker (which spans from Gly447 to Asp462) make numerous contacts with conserved residues of the GH9 module, emphasizing the importance of the linker in this interaction (Fig. 6). A conserved Gly447 of the linker interacts via hydrogen bonds with Gly444 and Tyr440 of the GH9 module. Pro449 forms hydrophobic interactions with Tyr440, and Asp450 forms hydrogen bonds with Gln11. Another linker residue, Phe453, forms hydrophobic and aromatic interactions with two aromatic residues of the GH9 module, i.e., Phe15 and Tyr399. Gly455 forms hydrogen bonds with Asn33 and Arg31. Glu457 forms intricate interactions with a variety of residues of the GH9 module, which include hydrogen bonding with Asn33, Val397, Arg395, as well as hydrogen and ionic interactions with His396. Glu461 exhibits hydrogen and ionic interactions with Arg395. Additionally, linker residues Glu457 and Glu461 form hydrogen bonds and salt bridges with Trp490 and Lys487, respectively. The latter belong to the CBM3c module and are part of a loop (486–498), which protrudes towards and forms interactions with several amino acids of the GH9 module. There are also many hydrogen bonds formed between the neighboring amino acids of the linker, thus contributing to its defined conformation. Altogether the multiple, well-conserved interactions between the linker, the GH9 and the CBM3c modules stabilize the spatial orientation of the modules towards one another and contribute to the structural rigidity of the entire molecule, resulting in an active enzyme structure.

As mentioned above, the mutual spatial orientation of the GH9 and CBM3c modules is very similar to that in the native, intact bimodular pairs from Cel9G and Cel9A leading to the overall similarity in structures. The remarkable conservation of the overall architecture in the reassembled *in vitro* complex together with the striking conservation of the contact residues implies its high functional importance. In all of these structures (Cel9G, Cel9A, and the reassembled GH9-CBM3c*L* from Cel9I), the flat surface of the CBM3c module is aligned in continuation with the catalytic cleft of the GH9 module, making an extended platform (Fig. 4B). This platform is rich in charged and polar surface residues that are highly conserved throughout the family 3 CBMc's.

## Microcalorimetric analysis of the GH9-CBM3c complex formation

The binding constants of GH9 and the CBM3c were obtained by performing isothermal titration calorimetry (ITC) experiments in which a solution of GH9 was titrated with a solution of CBM3c with or without the linker (Fig. 7). Control experiments for each of the components alone were conducted and subtracted from the titration data. In both cases the titration curve could be fitted to a one-site binding model although the calculated stoichiometry was less than one. The low stoichiometry is probably a result of the fact that the soluble CBM module lost its functionality with time and its true active concentration was less than the measured protein concentration. To estimate the binding constants for the two CBM3c forms, the CBM3c concentrations were corrected to provide a stoichiometry of one. In all cases, the binding reactions were enthalpy driven with a negative entropy contribution. CBM3cL provided binding constants ($K_d$) between $1.3–2.0 \times 10^{-6}$ M, whereas CBM3cNL exhibited stronger binding constants, $K_d$ between

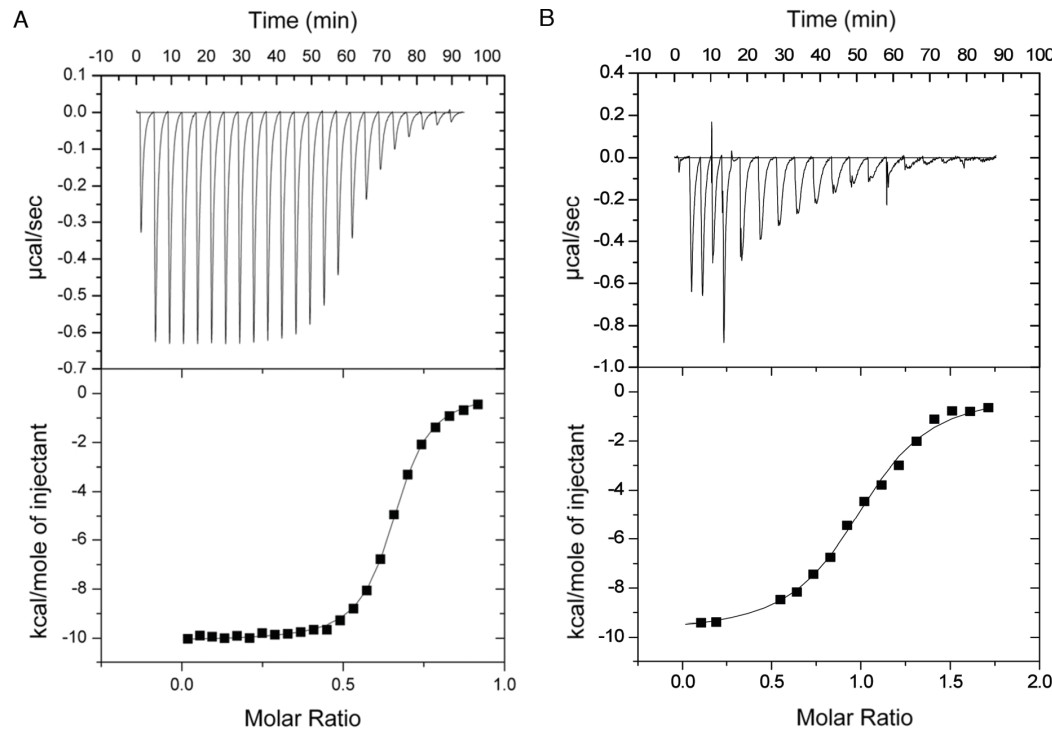

**Figure 7** **Representative ITC titration of (A) GH9 and CBM3c*NL* (B) GH9 and CBM3c*L*.** The top panel shows the calorimetric titration and the bottom panel displays the integrated injection heats corrected for control dilution heat. The solid line is the curve of the best fit used to derive the binding parameters, and the fitted data describe an interaction of a one binding site model.

$2.9–4.3 \times 10^{-7}$ M. Thus, the linker may serve as a mitigating factor for the binding process, ensuring specific binding orientation. This is consistent with the structural data and the activity assays, which emphasizes the important role of the linker in enzyme functioning. In the case of CBM3c*NL*, the binding process may occur faster in the absence of linker, but may also lead to unspecific binding and aggregation of the modules.

## DISCUSSION

A striking feature of the family 9 glycoside hydrolases is their subdivision into architectural themes, which are defined by their conserved modular composition (*Bayer, Shoham & Lamed, 2006*). In this context, the Theme B1 endoglucanases contain a GH9 catalytic module followed by a purportedly fused family 3c CBM. Biochemical studies of some of the members of this group (*Arai et al., 2001*; *Chiriac et al., 2010*; *Gal et al., 1997*; *Irwin et al., 1998*; *Li, Irwin & Wilson, 2007*) have shown that the CBM3c acts as a modulator of the function of the catalytic module. However, the exact manner in which the CBM3c functions is still unclear. It has been shown (*Gal et al., 1997*; *Gilad et al., 2003*; *Irwin et al., 1998*) that family 3c CBMs (including the CBM3c from *C. thermocellum* Cel9I) fail to bind insoluble cellulosic substrates, implying that they do not act as targeting agents for such substrates. The targeting of the enzyme to crystalline cellulose is achieved either

through an additional CBM (*Kostylev et al., 2012*) or by attachment of the enzyme to a CBM-containing scaffoldin via a cohesin-dockerin interaction (*Mingardon et al., 2011*).

The CBM3c module of Cel9A from the *T. fusca* has been proposed to loosely anchor the enzyme to cellulose, to disrupt the hydrogen bonds in crystalline cellulose and to guide a single cellulose strand towards the active site of the GH9 catalytic module (*Bayer, Shoham & Lamed, 2006*; *Li, Irwin & Wilson, 2007*). This hypothesis has been supported by molecular docking and molecular dynamics simulation studies (*Oliveira et al., 2009*). Moreover, double point mutations indicated that high coordination between the substrate affinities of the catalytic module and CBM needs to be precisely controlled (*Li, Irwin & Wilson, 2010*). Enzyme thermostability was reported to be affected by the presence of the CBM3c probably due to the formation of a compact structure (*Chiriac et al., 2010*; *Su, Mackie & Cann, 2012*; *Yi et al., 2013*).

The previously reported structures of Cel9A from *T. fusca* (*Sakon et al., 1997*) and Cel9G from *C. cellulolyticum* (*Mandelman et al., 2003*) revealed that the catalytic module and the CBM3c are separated by a ~20-residue linker that forms multiple polar and hydrophobic interactions mainly with the GH9 module. In an earlier report, we demonstrated that separately expressed GH9 and CBM3c*L* from Cel9I of *C. thermocellum* interact with one another to form an enzymatically active complex (*Burstein et al., 2009*). In the current article, we showed further that the GH9 and CBM3c can also be reassembled without the linker, albeit at the expense of catalytic activity, thus emphasizing the importance of the linker in positioning correctly the CBM relative to the GH9 catalytic module.

There is evidence that linkers in multi-modular proteins may serve communication roles between the modules via allosteric mechanisms and variation in their sequences affect enzyme activity (*Ma et al., 2011*). Linker length and rigidity was shown to play a critical role in the cooperative action of the catalytic module of a cellulase and a CBM (*Ting, Makarov & Wang, 2009*). Computational studies of *T. fusca* Cel9A suggested that thermal contributions to enzyme plasticity and molecular motion at high temperatures may play a role in enhancing CBM and catalytic domain synergy, and the linker may have an important role in this process (*Batista et al., 2011*). The length of the linkers may also play an important role in protein function and adaptation to the environment (*Sonan et al., 2007*). Studies in cellulolytic fungi revealed that linkers undergo modifications such as glycoslation and have also been shown to directly bind to the cellulose substrate (*Beckham et al., 2012*; *Payne et al., 2013*; *Sammond et al., 2012*; *Srisodsuk et al., 1993*). Point mutations in different fungal GH-CBM linkers have also been shown to significantly affect the activity of the enzymes and their stability (*Couturier et al., 2013*; *Lu et al., 2014*).

The characteristics of the reassembled linker-containing complex are corroborated by the X-ray crystallographic data. Indeed, it is quite surprising that the two separately expressed entities recombined in such a way that the complex could in fact be crystallized. Moreover, the resultant structure was remarkably similar to the known structures of the intact bimodular GH9-CBM3c pairs from *C. cellulolyticum* Cel9G and *T. fusca* Cel9A. Accordingly, the vast majority of the contact residues are similar among the three structures. Conserved residues of the linker make contacts with conserved residues of

the GH9 module, highlighting the importance of the linker in this interaction. Multiple hydrogen, ionic and hydrophobic bonds between the linker and the functional GH9 and CBM3c stabilize the spatial organization of the modules. The similarity of the reassembled and native intact structures is particularly intriguing, as it suggests that folding of the modular structures and emplacement of the linker during biosynthesis and intermodular recognition during complex formation are governed by the same interactions, which may have distinct functional consequences. In contrast to the GH9-CBM3c*L*, the re-associated GH9-CBM3c*NL* complex never crystallized, suggesting that the reassembly of the two modules in the absence of linker was somewhat heterogeneous in character.

Single proteins commonly fold into defined structures, wherein their N- and C-terminal ends are in relatively close proximity to one another. If we view the structures of the Theme B1 enzymes, it is evident that their individual modules, the GH9 catalytic module and the CBM3c, are consistent with this rule. The positions of the N- and C-termini of the Theme B1 catalytic module are similar to those of the other GH9 thematic members, including those of Theme A, which lack additional modules. Likewise, the N- and C-termini of CBM3c are essentially the same as all other members of the family 3 CBMs, regardless of their source (i.e., parent cellulase, scaffoldin, etc.). The evolutionary significance of this observation is that, originally, the functional relationship between the two modules was likely a more conventional one, whereby an ancestral CBM3 played a standard targeting role to deliver the GH9 catalytic module to its substrate. During the course of evolution, this relationship changed, and the precise positioning and fusion of a mutated CBM3 with a GH9 catalytic module served to modulate the activity characteristics of the latter. For this purpose, the flat surface of the CBM3c is aligned with the flat surface of the catalytic module, and the appropriate residues that interact with the single cellulose chain are thus aligned with the active site of the GH9 module. As a consequence, the two closely juxtaposed modules can be considered as a single functional entity. The functional positioning and fusion of the two modules, however, are at odds with the inherent locations of the termini of the two modules, such that the C-terminus of the catalytic module is very distant from the N-terminus of the CBM3c. Consequently, nature has provided a very distinctive type of conserved linker, which both connects the two modules and helps secure their required orientation for processive endoglucanase activity.

## CONCLUSIONS

Cellulase 9I (Cel9I), a non-cellulosomal family 9 processive endoglucanase from *Clostridium thermocellum*, which degrades crystalline cellulose phosphoric acid-swollen cellulose (PASC) and carboxymethyl cellulose (CMC), consists of a catalytic GH9 module followed by two family 3 carbohydrate-binding modules (CBMs): CBM3c and CBM3b, separated by linker regions. C-terminal CBM3b module, as a classic CBM3, is responsible for targeting the Cel9I enzyme to the planar surface of the crystalline cellulose. The CBM3c is crucial for the GH9 enzymatic activity. In this work, we investigated the interaction of separately expressed catalytic module and CBM3c either with or without the intermodular

linker in order to better understand the function of the CBM3c in the family-9 enzymes and the role of the linkers regions.

GH9 catalytic module and CBM3c were able to interact and reassemble both with and without the linker; however, the linker was essential for the endoglucanase catalytic activity. Surprisingly, we were able to crystallize these two separately expressed entities, meaning that their reassembly was very ordered and structurally homogeneous. The molecular structure of the GH9 and CBM3c with the linker region showed that they form a complex similar in structure to the intact tandem GH9-CBM3c modules of the orthologous endoglucanases Cel9G from *C. cellulolyticum* and Cel9A from *Thermobifida fusca*. The flat, conserved surface of the CBM3c module is aligned in continuation with the catalytic cleft of the GH9 module, presumably forming one functional entity, which binds to the planar surface of the cellulose. Conserved residues of the linker make contacts with conserved residues of the GH9 module, highlighting the importance of the linker in this interaction. Overall, our results demonstrate that the linker regions in the GH9/CBM3c endoglucanases are necessary to achieve the right spatial organization of the modules and for the fixation of the modules into functional enzymes.

## ACKNOWLEDGEMENTS

This article is dedicated to the memory of Professor Felix Frolow, who passed away on 29 August 2014. We thankfully acknowledge the ESRF for synchrotron beam time and staff scientists of the ID-29 beam line for their assistance.

### Funding

This research was supported by the Israel Science Foundation (ISF; Grant nos. 293/08 to FF and 1349/13 to EAB). Additional support was obtained by a grant (No. 24/11) issued to RL by The Sidney E. Frank Foundation through the ISF. A grant was awarded to EAB from the F. Warren Hellman Grant for Alternative Energy Research in Israel in support of alternative energy research in Israel administered by the Israel Strategic Alternative Energy Foundation (I-SAEF). This research was also supported by the establishment of an Israeli Center of Research Excellence (I-CORE Center No. 152/11, EAB) managed by the Israel Science Foundation, from the United States-Israel Binational Science Foundation (BSF), Jerusalem, Israel, by the Weizmann Institute of Science Alternative Energy Research Initiative (AERI) and the Helmsley Foundation, and a grant to EAB and RL from the Israel Ministry of Science (IMOS). EAB was also supported by the European Union, Area NMP.2013.1.1-2: Self-assembly of naturally occurring nanosystems: CellulosomePlus Project number: 604530 and an ERA-IB Consortium (EIB.12.022), acronym FiberFuel. EAB is the incumbent of The Maynard I. and Elaine Wishner Chair of Bio-organic Chemistry. The funders had no role in study design, data collection and analysis, decision to publish, or preparation of the manuscript.

## Grant Disclosures

The following grant information was disclosed by the authors:
Israel Science Foundation: 293/08, 1349/13, 24/11.
Israel Strategic Alternative Energy Foundation (I-SAEF).
Israeli Center of Research Excellence: 152/11.
United States-Israel Binational Science Foundation (BSF).
Institute of Science Alternative Energy Research Initiative (AERI).
Helmsley Foundation.
Israel Ministry of Science (IMOS).

## Competing Interests

Edward Bayer is an Academic Editor for PeerJ.

## Author Contributions

- Svetlana Petkun conceived and designed the experiments, performed the experiments, analyzed the data, wrote the paper, prepared figures and/or tables.
- Inna Rozman Grinberg analyzed the data, wrote the paper, reviewed drafts of the paper, prepared figures and/or tables.
- Raphael Lamed conceived and designed the experiments, analyzed the data, wrote the paper, reviewed drafts of the paper.
- Sadanari Jindou performed the experiments, contributed reagents/materials/analysis tools.
- Tal Burstein conceived and designed the experiments, performed the experiments.
- Oren Yaniv conceived and designed the experiments, performed the experiments, analyzed the data.
- Yuval Shoham conceived and designed the experiments, contributed reagents/materials/analysis tools.
- Linda J.W. Shimon analyzed the data, wrote the paper, reviewed drafts of the paper.
- Edward A. Bayer conceived and designed the experiments, analyzed the data, wrote the paper, prepared figures and/or tables, reviewed drafts of the paper.
- Felix Frolow conceived and designed the experiments, analyzed the data, wrote the paper, prepared figures and/or tables.

## Data Availability

Protein Data Bank, accession code 2XFG.

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
