# Peer review of "Reassembly and co-crystallization of a family 9 processive endoglucanase from its component parts: structural and functional significance of the intermodular linker"

_PeerJ, doi:10.7717/peerj.1126_

## Round 0.1 · original submission · Major Revisions

· Academic Editor

Major Revisions

Dear Ed,

Your paper has been reviewed by three experts in the field, all found it interesting. One reviewer has some concerns that need be addressed.
I look forward to receiving your revised MS very soon.

Best

George

Reviewer 1 ·

Basic reporting

There is something wrong in these sentences:
1.PEG3350 is lost in the sentence “The best crystals were obtained in 30% PEG, 0.2 M magnesium chloride and 0.1 M Hepes pH 7.5”.
2."Clostridium thermocellum" should not be underlined (References).

In figure captions, ABCD should be highlighted.

Experimental design

No comments

Validity of the findings

Why is there only the result of ITC titration of GH9 and CBM3cNL in figure7? In this paper it is said that “GH9 was titrated with a solution of CBM3c with or without the linker (Figure 7)”.

Additional comments

No comments

Reviewer 2 ·

Basic reporting

No comments

Experimental design

No comments

Validity of the findings

No comments

Additional comments

This is a solid structural study about the regulatory roles of the linker in G9/CBM3C endoglucanases. I suggest publication after the following minor issues are addressed.

1. There is a brief description of Clostridium thermocellum (line 62-64). Is this strain very special in the degradation of cellulose? Is it very valuable in the industrial application? The authors are suggested to have more introduction to show why they choose this strain for study.

2. The key point of the study is to delineate the effects of the linker between GH9 domain and CBM3cL domain on the activity of the GH9 domain. The authors prepared the GH9 and CBM3cL domains and reconstituted them together to solve the complex structure. Why the authors did not express and purify the GH9 domain, linker, and the CBM3cL domain as one polypeptide and determine structure? The authors might be not able to express and purify this fragment or could not obtain the crystals? It needs to be explained or discussed in the revised manuscript.

3. The novel point of the study is the elucidation of the roles of the linker in the regulation of GH9 domain activity. However, there are three figures to compare the structure of GH9 domain with its homologous structures, and the structure of CBM3cL with its homologous structures. There is only one figure (Figure 6) to show the linker and there is no detailed description between the linker and the domains. The authors are suggested to have more description and discussion in this part.
4. Line 228: “3350 and 4000. The best crystals were obtained in 30 % PEG” .What PEG did the authors use to grow the crystals?

Reviewer 3 ·

Basic reporting

We can classify the paper in two part. The fisrt one is to crystallize these two separate part (GH9 and CBM3c) expressed, but only get the crystal and result of the CBM3c with a linker and GH9, and the result is really the same with previous studies. The second part is to make the ITC experiments to show the differece between CBM3c with or without linker. The second part is meaningful to make a conclusion because it is a comparation.The theme of this paper has showed that the author wanted to find the effect of the linker. But they didn't get the crystal without the linker so they couldn't make a comparation between these two situations. As the result with linker is the same with previou studies, it misses the innovation. And they didn't do some important site mutations to show the importance of the linker. They gave some explanation about it that without the linker the complex will lose the stability so they could't get the crystals. But it still needs to be proved by SLS/DLS. Some interesting ideas can have a try, for instance if the linker with CBM3c has a digisted site when the complex has been formed, we can try to cut CBM3c ,what will be in this situation ?
In a word, I think the paper lose some important and innovaive results. So I suggest it to be revised. I hope more and innovative work will be presented next time.

Experimental design

No Comments

Validity of the findings

No Comments

Additional comments

No Comments

---

## Round 0.2 · accepted · Accept

· Academic Editor

Accept

Dear Ed,

I am pleased to inform you that your revised paper is accepted by PeerJ.

Best,

George
George Guo-Qiang CHEN